# Factors Influencing Disaster-Incident-Related Impacts on Korean Nursing Students

**DOI:** 10.3390/ijerph16245111

**Published:** 2019-12-14

**Authors:** Minkyung Gu, Ran Kim, Hyunjung Lee, Sohyune Sok

**Affiliations:** 1Department of Nursing, College of Science and Technology, Daejin University, Pocheon-si, Gyeonggi-do 11159, Korea; g-minkyung@hanmail.net; 2Department of Nursing, Graduate School, Kyung Hee University, Seoul 02447, Korea; knan9008@hanmail.net (R.K.); hunjung77@sd.go.kr (H.L.); 3College of Nursing Science, Kyung Hee University, 26, Kyungheedae-ro, Dongdaemun-gu, Seoul 02447, Korea

**Keywords:** disasters, health status, coping skill, nursing student

## Abstract

The frequency of earthquakes in South Korea is increasing. This study aimed to examine and identify the factors influencing the degree of disaster-incident-related impacts among Korean nursing students who have actual disaster experience. The study sample consisted of 153 nursing students living around the Phohang-si area in Gyeongsang-do, South Korea, and who have actual disaster-incident-related experience. Measures used in this study were the Impact of Event Scale, Perceived Health Status Scale, Psychological Well-Being Scale, and Coping Strategy Indicator (Korean version). The data collection period was from October to December 2018. Factors that influence disaster-incident-related impacts among Korean nursing students in descending order are as follows: perceived health status (β = 0.48), gender (β = −0.28), coping skill (β = 0.18), psychological well-being (β = 0.14), need for disaster education (β = 0.12), and major satisfaction (β = −0.12). This study provides preliminary evidence that perceived health status is a major and primary predictor of disaster-incident-related impacts among Korean nursing students, followed by coping skill and psychological well-being. The findings can be reflected in a pertinent curriculum by actively considering these factors in designing nursing education interventions for managing disaster-incident-related impacts among Korean nursing students.

## 1. Introduction

South Korea is situated on the eastern edge of the Eurasian Plate, which is widely recognized as an earthquake safety zone [1]. The magnitude 5.8-sized Gyeongju earthquake in September 2016, however, led to a shift in this perception, while the magnitude 5.4 earthquake that occurred in the Red Sea region, 6 km north of Pohang, in November 2017 raised the people’s awareness of the damage that can be caused by earthquakes [2,3]. Today, South Korea is constantly experiencing new types of natural disasters and accidents, and many regions in the country suffer damage from such disasters [1].

Earthquakes are natural phenomena that can cause severe damage to a country in terms of people’s lives and property, including related damages due to fires, collapses, explosions, traffic accidents, and environmental pollution [1]. The frequency of earthquakes in South Korea is increasing, along with extreme weather and global warming, due to the country’s geographical condition; particularly, the fact that it is located between continental and marine climates [4]. The occurrence of an earthquake may pose enormous obstacles to people’s perceived health status, psychological well-being, individual coping skills, and quality of life [5,6,7]. Socially, it may result in many casualties and property damage, which can negatively affect people’s living environment [4]. In particular, shock from disasters such as earthquakes may have a very negative impact on people’s physical, psychological, and social health, and may cause mental pain that can lead to serious adverse effects in terms of adjusting to reality [5,8]. Currently, there is an active debate in South Korea about the need for an advanced disaster management system in preparation for earthquakes [1,4]. There has also been a social demand for establishing such a system [1].

Disaster preparedness education, which is being implemented in South Korea, has been conducted in the form of generic events by national agencies, so it is not appropriate as an educational method for enhancing the disaster preparedness of individuals. For this reason, disaster-related courses are still operated as elective courses instead of requirements in emergency nursing education. However, in the United States, in the wake of the September 11 terrorist attacks, nursing colleges are known to operate separate courses for disaster nurses. However, the Korean Academy of Disaster Nursing was not established until 2012 in South Korea, and there are still only a small number of disaster nursing education courses, which are mainly for nurses and medical workers. Disaster nursing has only recently been included in the nursing education curriculum, so the need to standardize disaster preparedness education content should be a top priority. However, the discussion of the operational effectiveness of disaster education against earthquakes has empirical difficulties because it should be approached as educational policies related to current disaster preparedness operation. At present, there is an active debate about the need for an advanced disaster management system in preparation for earthquakes. It also addresses the importance of establishing an effective disaster management system against earthquakes in response to such social demands.

The recently studied prior literature on nursing students in relation to disasters included a study on the degree of disaster nursing awareness of nursing students; disaster nursing competence and self-resilience; disaster preparedness; core performance and educational needs of disaster nursing; and disaster experience, disaster awareness, and stress related to perceived disaster [9,10,11,12,13,14]. Few studies [12,14] have been conducted on nursing students who have actually experienced disaster-incident-related cases. If enhanced theoretical and practical courses are provided in the nursing education curriculum based on the results of research on nursing students who have actual experience, the disaster nursing competency of nursing students and nurses will be improved.

Moreover, the factors related to the impact and pain of nurses in the face of disaster-related incidents are inaccurately known or misunderstood, which seems attributable to an inappropriate disaster-related curriculum. Therefore, it is necessary to explore the factors related to impact in order to proceed with disaster-related courses more effectively.

Therefore, this study was conducted to investigate the degree of impact from a disaster incident and the study variables related to disaster-incident-related impacts experienced by nursing students. Factors that influence disaster-incident-related impacts are subsequently analyzed. Furthermore, this study highlighted the necessity of nursing education and preparation for coping with disasters, and ultimately aimed to come up with the basic data necessary for disaster nursing education.

The purpose of this study was to examine and identify the factors influencing the degree of disaster-incident-related impact on nursing students. The aims of the study were as follows: (1) to identify the general characteristics of nursing students who have experienced disaster-incident-related cases; (2) to examine the degree of disaster-incident-related impact and factors related to it; (3) to examine the degrees of disaster-incident-related impact according to the general characteristics of the study participants; (4) to examine the correlations between the degree of disaster-incident-related impact and the factors related to it; and (5) to determine the factors that influence the degree of disaster-incident-related impact.

## 2. Materials and Methods

### 2.1. Study Design

A cross-sectional descriptive design was employed to examine and identify the factors influencing the degree of disaster-incident-related impact on Korean nursing students.

### 2.2. Setting and Participants

The participants were made up of a total of 153 nursing students who have been living around Phohang-si in Gyeongsang-do, South Korea. The area around Phohang-si, Gyeongsang-do, South Korea was where the earthquakes occurred. Participants were recruited by convenient sampling. The eligibility criteria included an age of 20 years or older, attending a nursing school in the Phohang-si area, having experienced disaster-incident-related cases, an awareness of the purpose of this study, voluntary participation in the study, and having complete verbal communication ability in Korean. Exclusion criteria were nursing students who had no experience of disasters such as earthquakes. Two hundred nursing students met the inclusion criteria, and a total of 155 participated in this study. Among 155 questionnaires, 153 (98.71%) were collected. In total, 153 questionnaires were accepted in the final dataset because all were complete. The sample size adequacy (*N* = 129) using F test, G power 3.1.2 analysis software was estimated based on an alpha level 0.05, medium effect size 0.15, and power 0.95 [15]. Therefore, the sample size was adequate.

### 2.3. Data Collection

A nursing school in the area around Phohang-si, Gyeongsang-do, South Korea—where the earthquakes occurred—was visited for this study. Researchers contacted prospective study participants and discussed with them the purpose of this study, as well as the details of their participation and the questionnaire that would be used in data gathering. The questionnaires were given only to nursing students who sent their consent forms. The completed questionnaires were then returned from the nursing students. The survey was a self-reporting questionnaire that was administered by the researchers. The average time to complete the survey was 15~20 min. The data collection period was from October to December 2018.

#### 2.3.1. Primary Outcome

The Impact of Event Scale (IES) developed by Horowitz et al. [16] was revised and implemented to a Korean version by Eun et al. [17]. This scale was used to determine the degree of disaster-incident-related impact on the participants. It consists of a total of 22 questions on the emotions, feelings (Cronbach’s α = 0.752), and actions that remind one (Cronbach’s α = 0.820) of a disaster incident; the sleep disorders (Cronbach’s α = 0.748) and social constraints (Cronbach’s α = 0.763) that one developed due to a disaster incident; and one’s physical reactions to a disaster incident (Cronbach’s α = 0.825). This scale has five subscales based on the five-point Likert scale. The most likely score range would be 22–110 using subscales, and the higher the respondent’s score was, the higher the corresponding degree of disaster-incident-related impact. The reliability of the scale used in this study was based on a Cronbach’s α of 0.871.

#### 2.3.2. Covariates

The general characteristics of study participants consisted of a total of six items: gender, age, religion, need for disaster education, living together, and major satisfaction.

The perceived health status scale developed by Ware [18] and Speake et al. [19] was translated into Korean by Cho [20]. This scale was utilized to quantify the degree of the perceived health status of participants. It consists of three questions on one’s current health status, daily life disturbance, and health status compared to the other people in the same age group. It uses a five-point Likert scale. The probable score range would be 3–15, and the higher the respondent’s score was, the higher the degree of perceived health status. The reliability was based on a Cronbach’s α of 0.831.

The Korean version of Psychological Well-Being Scale (PWS) originally developed by Ryff and Keyes [21] was translated by Kim and Kim [22]. This scale was used to measure psychological well-being. It consists of autonomy (Cronbach’s α = 0.739), environmental control (Cronbach’s α = 0.746), personal growth (Cronbach’s α = 0.791), life purpose (Cronbach’s α = 0.710), positive interpersonal relationship (Cronbach’s α = 0.831), and self-acceptance (Cronbach’s α = 0.771). This scale consists of a total of 45 questions using a five-point Likert scale. Possible scores are 45 to 225, and the higher the respondent’s score was, the greater the level of psychological well-being. The reliability of the scale was based on a Cronbach’s α of 0.801.

The Coping Strategy Indicator (CSI) developed by Amirkan [23] was translated into Korean by Shin and Kim [24]. This scale was used to measure the degree of coping skill. This scale consists of the three sub-dimensions of seeking social support (Cronbach’s α = 0.885), problem-solving-centeredness (Cronbach’s α = 0.923), and avoidance-centeredness (Cronbach’s α = 0.931). This scale included a total of 33 questions using a five-point Likert scale. Possible scores are 33 to 165, and the higher the respondent’s score was, the higher the level of coping skill. The reliability of the scale was based on a Cronbach’s α of 0.975.

### 2.4. Statistical Analysis

The collected data were analyzed with the SPSS version 21.0 statistical software program. The general characteristics of the study participants and the levels of study variables were determined by descriptive statistics using frequency, percentage, mean, and standard deviation. The degrees of disaster-incident-related impact according to the general characteristics of the study participants were analyzed by *t*-test, F test, and post hoc test. Correlations among study variables related to disaster-incident-related impacts were probed by using the Pearson correlation coefficient. In the course of investigating the factors that influenced levels of disaster-incident-related impacts, multiple regression analysis was used.

### 2.5. Ethical Approval

The Institutional Review Board of the university in Gyeonggi-do, South Korea approved this study (IRB No. 1040656-201810-SB-01-05) in terms of ethical considerations. Participants were briefed that their involvement in the study would be voluntary, there was no penalty for not participating in this study, and they could withdraw their commitment at any time. They were also told that the information they would give would remain anonymous and confidential. The researchers then obtained completed written consent forms from the study participants.

## 3. Results

Table 1 shows the general characteristics of study participants. Most of the participants were female (73.2%), with males accounting for 26.8%. The average age of the participants was 23.33 years old, and the largest age group was 23–25 years old (52.9%). In terms of need for disaster education, 52.9% of the participants responded “Yes”. Most of the participants were living with family (71.2%), followed by living alone in a dormitory (25.5%). As for exercising, 51.7% of participants were exercising 1–2 days/a week. As for major satisfaction, 51.6% of the participants were satisfied, and 41.8% were moderately satisfied. Also, most of the participants had morning stiffness (84.1%) and had other diseases (79.6%).

In Table 2, the mean score of disaster-incident-related impact was 74.26 (SD = 12.99), indicating a higher level of disaster-incident-related impact compared to the median value score of 66. Their mean score in terms of perceived health status was 10.28 (SD = 2.09), showing a slightly higher level of perceived health status compared to the median value score of 9. The participants’ mean score in terms of psychological well-being was 111.71 (SD = 11.37), exhibiting a lower level of psychological well-being compared to the median value score of 135. The mean score of the participants in terms of coping skill was 104.54 (SD = 24.95), demonstrating a higher level of coping skill compared to the median value score of 99.

In terms of the degrees of disaster-incident-related impact according to the general characteristics of the study participants, there were significant differences by gender (t = 3.341, *p* < 0.001), age (F = 2.117, *p* = 0.001), religion (t = 1.669, *p* = 0.016), and major satisfaction (F = 3.199, *p* < 0.001), as shown in Table 3.

In the correlations between the participants’ disaster-incident-related impact and related factors, the analyses of the participants’ perceived health status (r = −0.699, *p* < 0.001), psychological well-being (r = −0.357, *p* < 0.001), and coping skill (r = −0.469, *p* < 0.001) showed negative correlations, as shown in Table 4.

Table 5 depicts the results of multivariate regression presenting factors influencing disaster-incident-related impact. All assumptions of the regression analysis in this study coincided with the required assumptions of the regression equations. There was no multicollinearity problems (Durbin–Watson value = 1.685; tolerance limit = 0.243~0.765; Variance Inflation Factor (VIF) = 1.108~1.665). All the study variables were established to be independent of one another (correlations among study variables: from −0.699 to 0.392). For the assumption of the linearity model, the normality of the error term and homoscedasticity were satisfactory.

Multiple-regression analyses of the participants’ perceived health status, psychological well-being, and coping skill, as well as of the general characteristics of the participants’ gender, age, religion, need for disaster education, living together, and major satisfaction, were conducted to identify the key factors behind the level of disaster-incident-related impact. The analyses proved that the prediction model for the levels of disaster-incident-related impacts of Korean nursing students was significant (F = 24.46, *p* < 0.001). The value of the adjusted R^2^ was 0.607, corresponding to an explanatory power amounting to 60.7%. The perceived health status was found to be most influential on the disaster-incident-related impact of Korean nursing students (β = 0.48), followed by gender (β = −0.28), coping skill (β = 0.18), psychological well-being (β = 0.14), need for disaster education (β = 0.12), and major satisfaction (β = −0.12).

## 4. Discussion

In the present study, the disaster-incident-related impacts experienced by Korean nursing students were relatively higher than the median value. Although we found no studies using the same measurement scale, the result is similar to the study results obtained by Ahn and Kim [4] and Woo et al. [25] on nursing students, which reported 18.6 out of 25 points in their perception of disaster, and 3.71 out of 5 points in their awareness of disaster, respectively. This seems attributable to the dominant perception that South Korea is a relatively safe zone for earthquakes. This can be interpreted as the result of the tendency to perceive a disaster such as an earthquake as a sight to behold, and also of the mindset of safety insensitivity. As such, it is necessary to change nursing students’ perception of disasters in the future. To change their perception and to promote prompt response to disasters, it is necessary to prepare an efficient disaster response education program and make it a requirement for all South Koreans.

The female students in this study yielded a higher degree of disaster-incident-related impact than the male students. In the study of Ahn and Kim [4], the male students with military service experience had some training related to disasters, so it could be predicted that the degree of their disaster-incident-related impact was relatively low. Therefore, when operating a disaster nursing course particularly related to earthquakes, a concrete and systematic disaster education method particularly pertaining to earthquakes should be designed and operated to strengthen the simulation training related to disasters [26,27]. This would allow the trainees to experience real disaster situations by using audiovisual materials, in addition to theoretical education, and enhancing the trainees’ disaster coping skills [4,26].

In the present study, the lower the age of a nursing student, the higher the degree of disaster-incident-related impact. In the study of Woo et al. [25], those in the lower grades did not take a course related to disasters, so it was more difficult to predict their disaster-incident-related impact. In real-life disasters such as earthquakes, anyone can become a major and direct victim, regardless of age. Thus, a prompt realization of the need for disaster nursing education is necessary so that preparations can be done for disaster incidents including earthquakes, and to foster disaster preparedness [28]. Providing an efficient disaster response curriculum should also be prioritized through the operation of disaster nursing courses required for the lower grades and establishing continuity among and linking all the existing disaster nursing education programs for individuals, schools, workplaces, and communities [25].

The correlations between disaster-incident-related impact and the variables in this study showed that the higher the perceived health status, psychological well-being, and coping skills of a nursing student are, the lower the disaster-incident-related impact. The results were similar to those of the studies of Schmidt et al. [29] and Su et al. [30] on medical students, which reported that psychological instability due to disaster-incident-related impact leads to a high stress level, which causes excessive tension and anxiety, making it difficult for one to efficiently cope with disasters such as earthquakes. Therefore, efforts are needed to help nursing students form an active self-concept and to reinforce their psychological disaster preparedness to enable them to positively respond to disaster incidents [28].

In the present study, the most important factor affecting disaster-incident-related impacts on nursing students was perceived health status, followed by gender, coping skills, psychological well-being, the need for disaster education, and major satisfaction. It is difficult to directly compare these results because there have been few studies on Korean nursing students who have experienced disasters such as earthquakes. The results of this study, however, are similar to those of the study of Kim [13] on nursing students, which reported that, when there is no disaster experience, the ability to cope with disasters decreases, and productive and integrative activities are important to reduce the degree of disaster-incident-related impact on nursing students through the formation of various social and psychological relationships. In addition, the results of the present study support the findings of Schmidt et al. [29] that reveal disaster-like situations tend to make one less psychologically unstable. The more emotionally stable one is, the less the tendency to experience stress when a disaster occurs, thereby leading to a relatively good coping ability. To reduce the degree of disaster-incident-related impact, a disaster preparedness education intervention plan can be formulated with a focus on the psychological aspect. With this focus, one’s anxiety or tension in an earthquake-related situation can be reduced, ultimately reducing one’s stress. In addition, Haraoka et al. [28], Jose and Dufrane [27], and Lee et al. [11] mentioned the necessity of developing a systematic disaster education method to increase the disaster preparedness of nursing students. Thus, further research should be conducted on the various predictors affecting disaster-incident-related impact, which can be based on efficient disaster nursing education programs for nursing students.

The results of this study can be used to develop a broader understanding of the disaster nursing competency of Korean nursing students towards improvement, and can be reflected in the pertinent curriculum by actively considering these factors in designing nursing education interventions for disaster-incident-related impact for Korean nursing students. Additionally, in order to reduce the disaster-related impact on nurses who need to provide disaster nursing, the elements and factors that can reduce disaster-related impact should be included in the nursing education curriculum so that they can understand and recognize that it is only possible when their perceived health status is good. Above all, the need for regular disaster-related education for female nursing students who are relatively vulnerable should be perceived, and the nursing education curriculum should be improved so that they have time and opportunity to fully acquire the skills in clinical practice education, thereby improving confidence in their knowledge and competencies regarding disaster preparedness. Furthermore, they can be used to develop professional and skills-based disaster nursing programs and to cultivate competent nurses with expertise in disaster nursing. For Korean nursing students to complete an effective disaster-related nursing education program, it is necessary to develop a tool for measuring the disaster-related knowledge and practical disaster nursing competency of nursing students and to conduct experimental research to verify its effect by applying it. Gender analysis could perhaps be recommended, because sometimes the results in terms of health and well-being issues as well as the coping methods of men and women yield significant differences.

Disasters such as earthquakes can occur anywhere in South Korea. Thus, the paradigm of disaster management should focus on strengthening the community’s disaster resilience, enhancing preventive management, and promoting disaster prevention education and research in nursing colleges in South Korea.

Our study has several limitations. As there have been few studies related to the disaster-incident-related impact among Korean nursing students, care must be exercised when expanding the results of this study to explain the disaster-incident-related impacts experienced by all nursing students. In particular, as the study’s sampling was limited to the region where an earthquake had occurred, the results also have limitations in explaining the disaster-incident-related impacts experienced by all nursing students. As such, it is necessary to repeat and expand the study for the purpose of generalizing the factors affecting the disaster-incident-related impacts experienced by nursing students.

## 5. Conclusions

Based on the results of this study, the factors affecting the degree of disaster-incident-related impacts experienced by Korean nursing students were found to be perceived health status, coping skills for disaster-incident-related impacts, and psychological well-being. In other words, to reduce the degree of disaster-incident-related impact experienced by Korean nursing students, it is necessary to improve their perceived health status and psychological well-being as well as to reinforce their way of positively coping with disaster-incident-related impacts. Furthermore, to develop a systematic curriculum for the improvement of Korean nursing students’ disaster coping ability and to minimize such an impact, modules for properly preparing the country’s nursing students for earthquake-related situations should be included in their education while also taking into account the grade level. The study also suggests that education courses should be specially recommended for women and younger students. Moreover, simulation practice is important in providing nursing students with a specific disaster experience. By doing so, the disaster nursing competency of nursing students and nurses will be improved.

## Figures and Tables

**Table 1 ijerph-16-05111-t001:** General characteristics of the study participants.

Characteristics	Categories	*n*	%
Gender	Male	41	26.8
	Female	112	73.2
Age (year)	20~22	59	38.6
	23~25	81	52.9
	26~28	13	8.5
Religion	Yes	81	52.9
	No	72	47.1
Need for disaster education	Yes	81	52.9
	No	72	47.1
Living together	With family	109	71.2
	Living alone (dormitory)	39	25.5
	Others	5	3.3
Major satisfaction	Satisfied	79	51.6
	Moderate	64	41.8
	Dissatisfied	10	6.6

**Table 2 ijerph-16-05111-t002:** Levels of disaster-incident-related impact, perceived health status, psychological well-being, and coping skill.

Variables	Range	Min	Max	Mean (SD)
Disaster-incident-related impact	22–110	40.00	110.00	74.26 (12.99)
Perceived health status	3–15	5.00	15.00	10.28 (2.09)
Psychological well-being	45–225	89.00	145.00	111.71 (11.37)
Coping skill	33–165	60.00	165.00	104.54 (24.95)

**Table 3 ijerph-16-05111-t003:** Degrees of disaster-incident-related impact according to the general characteristics of the study participants.

Characteristics	Mean	SD	t or F	*p*-Value	Scheffe
Gender			3.341	<0.001	
Male	82.78	14.70			
Female	71.14	10.81			
Age (year)			2.117	0.001	a, b > c
20~22	77.63	4.92 a			
23~25	77.59	13.55 b			
26~28	68.38	5.89 c			
Religion			1.669	0.016	
Yes	73.03	12.20			
No	75.63	13.79			
Need for disaster education			0.863	0.708	
Yes	72.66	14.25			
No	76.05	11.25			
Living together with family	74.13	13.13	1.080	0.366	
Living alone (dormitory)	75.61	12.33			
Others	66.40	14.90			
Major satisfaction			3.199	<0.001	a > b > c
Satisfied	77.65	11.88 a			
Moderate	71.37	13.75 b			
Dissatisfied	65.90	8.14 c			

**Table 4 ijerph-16-05111-t004:** Correlations between disaster-incident-related impact and the study variables.

Variables	Disaster-Incident-Related Impact	Perceived Health Status	Psychological Well-Being	Coping Skill
	r (*p*-Value)			
Disaster-incident-related impact	1			
Perceived health status	−0.699 (<0.001)	1		
Psychological well-being	−0.357 (<0.001)	0.241 (0.003)	1	
Coping skill	−0.469 (<0.001)	0.392 (<0.001)	0.240 (0.003)	1

**Table 5 ijerph-16-05111-t005:** Factors influencing disaster-incident-related impact.

Variables	Model 1						
	B	S.E	β	t	*p-*Value	95% CI	
						Lower	Upper
Gender	−23.18	3.23	−0.79	−7.17	<0.001	−29.569	−16.797
Age (year)	−3.88	0.93	−0.44	−4.14	<0.001	−5.744	−2.035
Religion	0.29	1.82	0.01	0.16	0.871	−3.312	3.907
Need for disaster education	4.40	1.77	0.12	1.76	0.080	−1.249	10.049
Living together	−0.50	1.63	−0.02	−0.30	0.761	−3.735	2.736
Major satisfaction	−7.10	1.44	−0.33	−4.92	<0.001	−9.952	−4.248
R2 = 0.361, Adjusted R2 = 0.330, F(*p-*value) = 11.70 (<0.001)							
**Variables**	**Model 2**						
	**B**	**S.E**	**β**	**t**	***p-*Value**	**95% CI**	
						**Lower**	**Upper**
Gender	−8.31	2.90	−0.28	−2.86	0.005	−14.043	-2.578
Age (year)	−0.97	0.78	−0.11	−1.24	0.217	−2.538	0.581
Religion	2.00	1.42	0.07	1.40	0.163	−.821	4.822
Need for disaster education	3.15	1.36	0.12	2.31	0.022	0.017	8.681
Living together	1.04	1.28	0.04	0.81	0.416	−1.488	3.578
Major satisfaction	−2.52	1.19	−0.12	−2.10	0.037	−4.885	−0.158
Perceived health status	3.00	0.39	0.48	7.63	<0.001	2.229	3.786
Psychological well-being	0.16	0.06	0.14	2.73	0.007	0.047	0.292
Coping skill	0.09	0.03	0.18	3.17	0.002	0.037	0.160
R2 = 0.633, Adjusted R2 = 0.607, F(*p-*value) = 24.46 (<0.001)							

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
