# Peer review of "Factors Influencing Disaster-Incident-Related Impacts on Korean Nursing Students"

_ijerph, 2019, doi:10.3390/ijerph16245111_

Round 1

Reviewer 1 Report

Good structure and writing. The content of study design is too brief.

Author Response

Reviewer 1

Good structure and writing. The content of study design is too brief.

=> Please see the red letters in 2.1. Study Design.

We authors added.

Reviewer 2 Report

This study explores factors associated with impact of Earthquakes in aa sample of Korean nursing students. While the study question is interesting, there are important changes that need to be made before it is suitable for publication.

Abstract and throughout the text

Revise English (“and who having the experience of disaster-incident-related shock”, “There has been an active debate in South Korea of late”, or “The most of gender was”).

Please use language that avoids assuming causality as this is not an experimental study (“The factor that was found to have the most influence on”, “factors that influenced their levels of disaster-incident-related shock”, or “the most important factor affecting”). This affects the title too.

How would the authors exactly use the results to design nursing education interventions if the most important factors associated with shock in the study were health status and gender?

Introduction

The introduction is not well linked with the study goals. The authors expose that the current courses for nursing students are not sufficient, not because they do not tap into the adequate outcomes, but because they are not mandatory. How does the present study solve this problem? The present study would be justified if the authors explained that there is a current misunderstanding of the factors associated with shock and distress in nurses, which has made current courses inadequate. Therefore, this would justify the need to explore factors that are actually associated with shock to guide courses in a more efficient way. This is not what the introduction is exposing.

Methods

How many participants failed to meet inclusion criteria and how many refused to participate?

Please use impact of event throughout the text instead of shock to be more consistent with the primary outcome. How many subscales does the IES have? It looks like 3 as per the authors’ description. Did the authors use subscales or a global score? Please describe and justify.

Results

Why do the authors use a global coping skill score if 3 subscales exist in the scale? Also, presenting one alpha score only would also be wrong in this case. This also applies to PWS.

Why do the authors use two blocks in the regression and why did they use such grouping? Please include beta CI in Table 5.

Discussion

Which median value do the authors refer to in the sentence “In the present study, the disaster-incident-related shock experienced by the Korean nursing students was relatively higher than the median value.” Please provide a reference study and describe the reference study and the population in such study.

Can the authors explain exactly why and how they would “change not only the nursing students’ but also all the other South Koreans’ perception of disasters like earthquakes in the future.”?

As suggested in the introduction, it is unclear what is not being done well in current courses for nursing students. The authors state that “Therefore, when operating a disaster nursing course particularly related to earthquakes, a concrete and systematic disaster education method particularly pertaining to earthquakes should be designed and operated to strengthen the simulation training related to disasters”. Is this not being done at the moment?

The authors refer to previous research showing that “psychological instability due to disaster-incident-related shock leads to a high stress level, which causes excessive tension and anxiety”. Well-being and health in this study were general scales and not specifically addressing the consequences of earthquakes, so the presented statement should be modified.

Conclusions

In the first paragraph, the authors ignore factors like age and gender as predictors of impact. Maybe the authors could use that to suggest that courses should be specially recommended for women and younger students. Additionally, the recommendations given for future courses are very vague (i.e., how would the authors use their results with health status and well-being to change future courses exactly).

Author Response

Revision Comments

This study explores factors associated with impact of Earthquakes in aa sample of Korean nursing students. While the study question is interesting, there are important changes that need to be made before it is suitable for publication.

Abstract and throughout the text

Revise English (“and who having the experience of disaster-incident-related shock”, “There has been an active debate in South Korea of late”, or “The most of gender was”).

==> Please see the red letters in Abstract and throughout the text.

We authors amended them.

Please use language that avoids assuming causality as this is not an experimental study (“The factor that was found to have the most influence on”, “factors that influenced their levels of disaster-incident-related shock”, or “the most important factor affecting”). This affects the title too.

==> Please see the red letters in Abstract, Title, and throughout the text.

We authors amended them.

How would the authors exactly use the results to design nursing education interventions if the most important factors associated with shock in the study were health status and gender?

==> Please see the red letters in Introduction, Discussion, and Conclusion.

We authors added and amended them.

Introduction

The introduction is not well linked with the study goals. The authors expose that the current courses for nursing students are not sufficient, not because they do not tap into the adequate outcomes, but because they are not mandatory. How does the present study solve this problem? The present study would be justified if the authors explained that there is a current misunderstanding of the factors associated with shock and distress in nurses, which has made current courses inadequate. Therefore, this would justify the need to explore factors that are actually associated with shock to guide courses in a more efficient way. This is not what the introduction is exposing.

==> Please see the red letters in Introduction.

We authors added and described.

Methods

How many participants failed to meet inclusion criteria and how many refused to participate?

==> Please see the red letters in 2.2. Setting and Participants.

We authors added and described.

Please use impact of event throughout the text instead of shock to be more consistent with the primary outcome.

==> Please see the red letters throughout the text.

We authors amended them.

How many subscales does the IES have? It looks like 3 as per the authors’ description. Did the authors use subscales or a global score? Please describe and justify.

==> Please see the red letters in 2.3.1. Primary Outcome.

We authors added and described.

Results

Why do the authors use a global coping skill score if 3 subscales exist in the scale? Also, presenting one alpha score only would also be wrong in this case. This also applies to PWS.

==> Please see the red letters in 2.3.2. Covariates.

We authors added and described.

Why do the authors use two blocks in the regression and why did they use such grouping?

==> The factors affecting disaster-incident-related impact were initially identified among the general characteristics of nursing students who actually experienced disaster-incident-related cases. After this initial analysis, the perceived health status, psychological well-being, and related coping strategies of nursing students with actual experience of disaster-incident-related shock were added in order to identify the factors affecting disaster-incident-related impact.

Please include beta CI in Table 5.

==> Please see the red letters in Table 5.

We authors added

Discussion

Which median value do the authors refer to in the sentence “In the present study, the disaster-incident-related shock experienced by the Korean nursing students was relatively higher than the median value.” Please provide a reference study and describe the reference study and the population in such study.

==> Please see the red letters in Discussion.

We authors added.

Can the authors explain exactly why and how they would “change not only the nursing students’ but also all the other South Koreans’ perception of disasters like earthquakes in the future.”?

==> Please see the red letters in Discussion.

We authors amended and added.

As suggested in the introduction, it is unclear what is not being done well in current courses for nursing students. The authors state that “Therefore, when operating a disaster nursing course particularly related to earthquakes, a concrete and systematic disaster education method particularly pertaining to earthquakes should be designed and operated to strengthen the simulation training related to disasters”. Is this not being done at the moment?

==> Please see the red letters throughout the text.

We authors amended and added.

The authors refer to previous research showing that “psychological instability due to disaster-incident-related shock leads to a high stress level, which causes excessive tension and anxiety”. Well-being and health in this study were general scales and not specifically addressing the consequences of earthquakes, so the presented statement should be modified.

==> Please see the red letters in Discussion.

We authors amended.

Conclusions

In the first paragraph, the authors ignore factors like age and gender as predictors of impact. Maybe the authors could use that to suggest that courses should be specially recommended for women and younger students.

==> Please see the red letters in Conclusion.

We authors added.

Additionally, the recommendations given for future courses are very vague (i.e., how would the authors use their results with health status and well-being to change future courses exactly).

==> Please see the red letters later in the Discussion.

We authors added.

Reviewer 3 Report

Dear authors,

It seems to me that you addressed a question of interest and that you evaluated relevant aspects to think about future interventions. However, I think this manuscript has been improved for publication.
The theoretical part does not sufficiently support the study proposed, what is known about the different variables and how they relate to each other.More literature and better explained is necessary to understand the work.
I also believe that the method could be reinforced. For example, I consider that, although the requirement is to have experienced an earthquake, when?, Is this variable controlled?
Regarding the results section, we know that, sometimes the results in health and well-being issues as well as the ways of coping with men and women pose differences, why has the regression analysis not been done separately?
Finally, I think that a matter of special interest in this work has to do with the possibility of thinking about future interventions. However, these are barely developed. I think it is necessary to develop them more, as well as the very meaning of this work and what, specifically, contributes to the field of knowledge.

Kind regards.

Author Response

Revision Comments

Comments and Suggestions for Authors

Dear authors,

It seems to me that you addressed a question of interest and that you evaluated relevant aspects to think about future interventions. However, I think this manuscript has been improved for publication.
The theoretical part does not sufficiently support the study proposed, what is known about the different variables and how they relate to each other. More literature and better explained is necessary to understand the work.

==> Please see the red letters in Introduction.

We authors added and amended.

I also believe that the method could be reinforced. For example, I consider that, although the requirement is to have experienced an earthquake, when?, Is this variable controlled?

==> Please see the red letters in Methods, Introduction and Discussion.

We authors added and amended.

Regarding the results section, we know that, sometimes the results in health and well-being issues as well as the ways of coping with men and women pose differences, why has the regression analysis not been done separately?

==> Please see the red letters later in Discussion.

We authors added.

Finally, I think that a matter of special interest in this work has to do with the possibility of thinking about future interventions. However, these are barely developed. I think it is necessary to develop them more, as well as the very meaning of this work and what, specifically, contributes to the field of knowledge.

Kind regards.

==> Please see the red letters in Discussion and Conclusion.

We authors added and amended.

Round 2

Reviewer 3 Report

Dear authors,

I really appreciate the effort to improve the manuscript.

Although this, I consider it would be needed reinforce the backgraound because althouth new information was included, no references were included. For example, the authors show that "no many2 studies have been developed regarding their main topic but which ones studied? Any reference to this, for example.

I consider the manuscript could be improved in same topics i showed first time but i consider they worked enough on this.

Best.

Author Response

Revision Comments

Reviewer 3 (Round 2)

Dear authors,

 I really appreciate the effort to improve the manuscript.

Although this, I consider it would be needed reinforce the background because although new information was included, no references were included. For example, the authors show that "no many2 studies have been developed regarding their main topic but which ones studied? Any reference to this, for example.

I consider the manuscript could be improved in same topics i showed first time but i consider they worked enough on this.

Best.

==> Please see the red letters in 3 pages, Introduction.

We authors added the reference for few studies.
